# Analyzing Resilience in the Greater Yellowstone Ecosystem after the 1988 Wildfire in the Western U.S. Using Remote Sensing and Soil Database

**Hang Li** [1,*] 🄳 **, James H. Speer** [1] 🄳 **and Ichchha Thapa** [2]

1   Department of Earth and Environmental Systems, Indiana State University, Terre Haute, IN 47809, USA;
    jim.speer@indstate.edu
2   Department of Forestry, Michigan State University, East Lansing, MI 48824, USA;thapaich@msu.edu
*   Correspondence: hli9@sycamores.indstate.edu

**Abstract:** The 1988 Yellowstone fire altered the structure of the local forest ecosystem and left large non-recovery areas. This study assessed the pre-fire drivers and post-fire characteristics of the recovery and non-recovery areas and examined possible reasons driving non-recovery of the areas post-fire disturbance. Non-recovery and recovery areas were sampled with 44,629 points and 77,501 points, from which attribute values related to topography, climate, and subsequent soil conditions were extracted. We calculated the 1988 Yellowstone fire burn thresholds using the differenced Normalized Burn Ratio (dNBR) and official fire maps. We used a burn severity map from the US Forest Service to calculate the burn severity values. Spatial regressions and Chi-Square tests were applied to determine the statistically significant characteristics of a lack of recovery. The non-recovery areas were found to cover 1005.25 km$^2$. Among 11 variables considered as potential factors driving recovery areas and 13 variables driving non-recovery areas, elevation and maximum temperature were found to have high Variance Inflation Factors (4.73 and 4.72). The results showed that non-recovery areas all experienced severe burns and were located at areas with steeper slopes (13.99°), more precipitation (871.73 mm), higher pre-fire vegetation density (NDVI = 0.38), higher bulk density (750.03 kg/m$^3$), lower soil organic matter (165.61 g/kg), and lower total nitrogen (60.97 mg/L). Chi-square analyses revealed statistically different pre-fire forest species ($p < 0.01$) and soil order ($p < 0.01$) in the recovery and non-recovery areas. Although Inceptisols dominated in both recovery and non-recovery areas, however, the composition of Mollisols was higher in the non-recovery areas (14%) compared to the recovery areas (11%). This indicated the ecological memory of the non-recovery site reverting to grassland post-disturbance. Unlike conventional studies only focusing on recovery areas, this study analyzed the non-recovery areas and found the key characteristics that make a landscape not resilient to the 1988 Yellowstone fire. The significant effects of elevation, precipitation, and soil pH on recovery may be significant to the forest management and forest resilience in the post-fire period.

**Keywords:** Yellowstone Fires of 1988; remote sensing; soil database; non-recovery areas; ecological resilience

## 1. Introduction

Disturbances followed by recovery shape forest ecosystems [1]. Global climate change triggers many ecological disturbances in the local forest, including fire, drought, and insect invasion, which challenge the current ecosystem structure [2–4]. If disturbance impact is within an acceptable range, the forest ecosystem will recover through its information legacy, which is a demonstration of resilience. This is an example of ecological adaptation to the past disturbances. If the disturbance impact goes beyond the range and breaks up the dynamic equilibria, information legacy and material legacy (the remains of past disturbances including surviving species, seed banks, nutrient pool, and undisturbed vegetation)

will provide the building blocks for the new forest ecosystem [5,6]. Understanding of the ecological memories consisting of information legacy (pre-fire drivers) and material legacy (post-fire characteristics) can help forest managers and researchers prepare for future forest disturbances [7,8]. We explored the influences of ecological memories (information legacy and material legacy) on fire recovery of the forest ecosystem.

The Yellowstone National Park was the first national park in the world, located in western U.S. The 1988 Yellowstone Fire is a good research area to examine the interactions between ecological disturbances and forest resilience [9]. The 1988 Yellowstone fire started in June 1988. A few ignitions occurred within a short period of time (Storm Creek Fire started on 14 June, Shoshone Fire started on 23 June, and other fires started between June and July). The fire finally damped down with snowfall on 11 September 1988. The 1988 Yellowstone fire was one of the largest and most significant fires in the United States [10]. This fire lasted more than three months, affected many forest areas, and largely changed the structure of the landscape in the Greater Yellowstone Ecosystem (GYE). The 1988 fire burned over 570,000 ha of the GYE. Christensen et al. [10] examined the reasons for the fire, such as long drought, wind, accumulated fuel, and some discrete fires caused by the lighting and human activities. Turner et al. [11] demonstrated that in the low severity areas, vegetation density returned to the prior-1988 fire level within two years. They also reported that char was found only in the topsoil from the surface to 14 mm depth, while the roots and rhizomes were still intact, even though the area was burned by crown fire in 1988. Romme et al. [12] believe that in the post-1988-fire period, appropriate temperatures and enough precipitation were important drivers for the recovery of the forest community. After 25 years of the recovery, spatial patterns and dominant tree species of the forest community in the post-fire period resemble those in the pre-fire period due to ecological memories. The forest recovered quickly in most areas.

Several studies focused on the recovery of the forest ecosystem following the 1988 Yellowstone fire [13–15]. However, few researchers analyzed the non-recovery areas, which are the areas that used to be forest in the pre-fire and turned to grassland or barren land in the post-fire. Due to the fire, roots could not keep soil stabilities, and holding water capacities in the forest ecosystem were largely impaired, resulting in some forest patches being converted to grassland or barren land [12]. Even though the areas of non-recovery were rare, the frequency of natural disasters, like landslides or mudslides, was higher than that in the recovery areas [16]. As an extreme case, the 1988 fire is a good example to analyze resilience and its impacts on forest management.

The objective of this study was to assess the characteristics of the recovery and non-recovery areas and examine the possible reasons for why few forest patches never recovered. This study hypothesized that the landscape characteristics (soil characteristics, pre-fire tree species, and topographic situation) were significantly different ($p < 0.05$) between non-recovery areas and recovery areas. Firstly, definitions of non-recovery areas and recovery areas were declared and their areas extracted using remote sensing images. Satellite images from 1987 to 2018 were selected to monitor the variations in vegetation density and burn severity. Secondly, information on local topographic, climate, vegetation density, and soil information as indicators of pre-fire drivers and post-fire characteristics were collected to find the possible drivers of non-recovery (Table 1).

**Table 1.** Pre-fire drivers and post-fire characteristics.

| General Factors | Specific Factors | Collected Data |
|---|---|---|
| Pre-fire drivers | Climate<br>Topographic<br>Biotic factors<br>Stable soil properties | t_mean, prec_a<br>DEM, slope<br>pre-fire NDVI, pre-fire tree species<br>soil order |
| Post-fire characteristics | Changeable soil properties | percent_sand, percent_silt, Mg,<br>OC, TN, pH, EC, BD |

Note: t_mean: mean temperature; prec_a: annual precipitation; OC: organic carbon; TN: Total Nitrogen; BD: bulk density.

## 2. Materials and Methods

### 2.1. Site Description

The GYE covers three states (Idaho, Wyoming, and Montana). Our study area is located at an elevation between 1455m to 3836m. The average annual temperatures in the summer and winter are 8 °C (46 °F) and –7 °C (20 °F), respectively. The average annual precipitation is 510 mm [17]. The Yellowstone wildfire in 1988 was the largest fire in this area based on historical records and tree-ring reconstructions [11,18]. The fire started from many small ignitions, and multiple separate fires occurred in June 1988. A long-term drought accumulated deadwood and dried materials, providing ample fuel. Once the blaze started, a dense understory served as a ladder fuel, which extended the fire to the crown of the pines [19,20], and the multiple fires joined to burn 36% of the National Park. The study area covering Yellowstone National Park and sections of the Shoshone National Forest are located within the areas between 109°49′22″ W to 111°19′22″ W longitude and 43°59′15″ N to 45°12′25″ N latitude, totaling 30,494.04 km² (Figure 1), which is most of one tile (Path: 38, Row: 29) in a Landsat image. Lodgepole pine (*Pinus contorta*) make up around 80% of the vegetation in the whole area. Other tree species included Engelmann spruce (*Picea engalmanii*), subalpine fir (*Abies lasiocarpa*), whitebark pine (*Pinus albicaulis*), and aspen (*Populus tremuloides*). The major soils in Yellowstone National Park are Inceptisols (45%), Mollisols (22%), Alfisols (19%), and other soils (14%) [21].

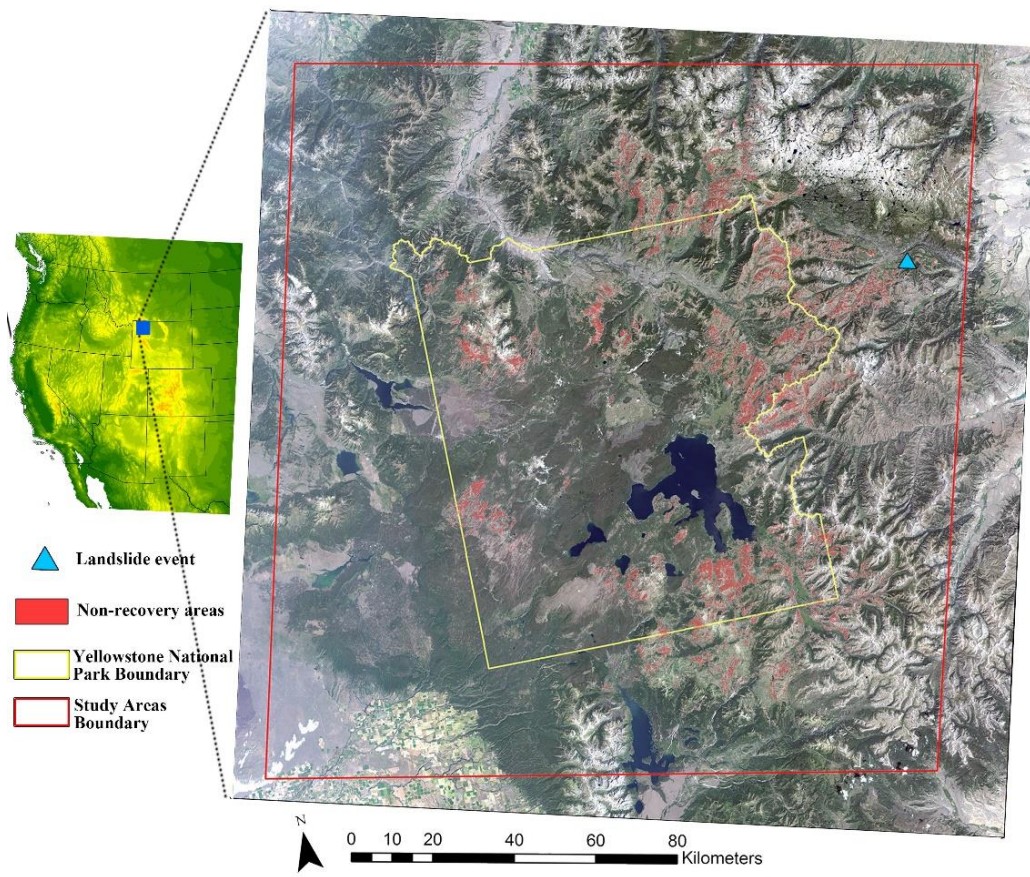

**Figure 1.** Study area dated 2 August 1989 after the 1988 wildfire.

*2.2. Data Collection*

In this study, all available Landsat series satellite images (30-m resolution) of five years (1987, 1989, 1998, 2008, and 2018) ranging from May to September were downloaded to avoid errors associated with changes in phenology (Figure 2). All satellite images were collected from the U.S. Geological Survey (USGS, https://earthexplorer.usgs.gov/ accessed on 18 May 2022). The best time to evaluate burned areas is during the growing season, especially in the summer. In the 1988 fire, the fire lasted the whole summer (until September), so we chose the images in 1989 to avoid the phenological influences and assess the burn severity accurately. Images with obscured pixels due to cloud and haze were removed. Each interval between the remaining temporal steps was around 10 years. Within each year, all images in the same year were overlaid, and median values were calculated, which could avoid the clouds and outliers in some extreme weather to generate a combined image. The combined 1987 image was used to measure the vegetation on the pre-fire landscape, and the difference with the 1989 image was used to measure fire severity. Other years' combined images were used to monitor how the forest recovered within each temporal interval.

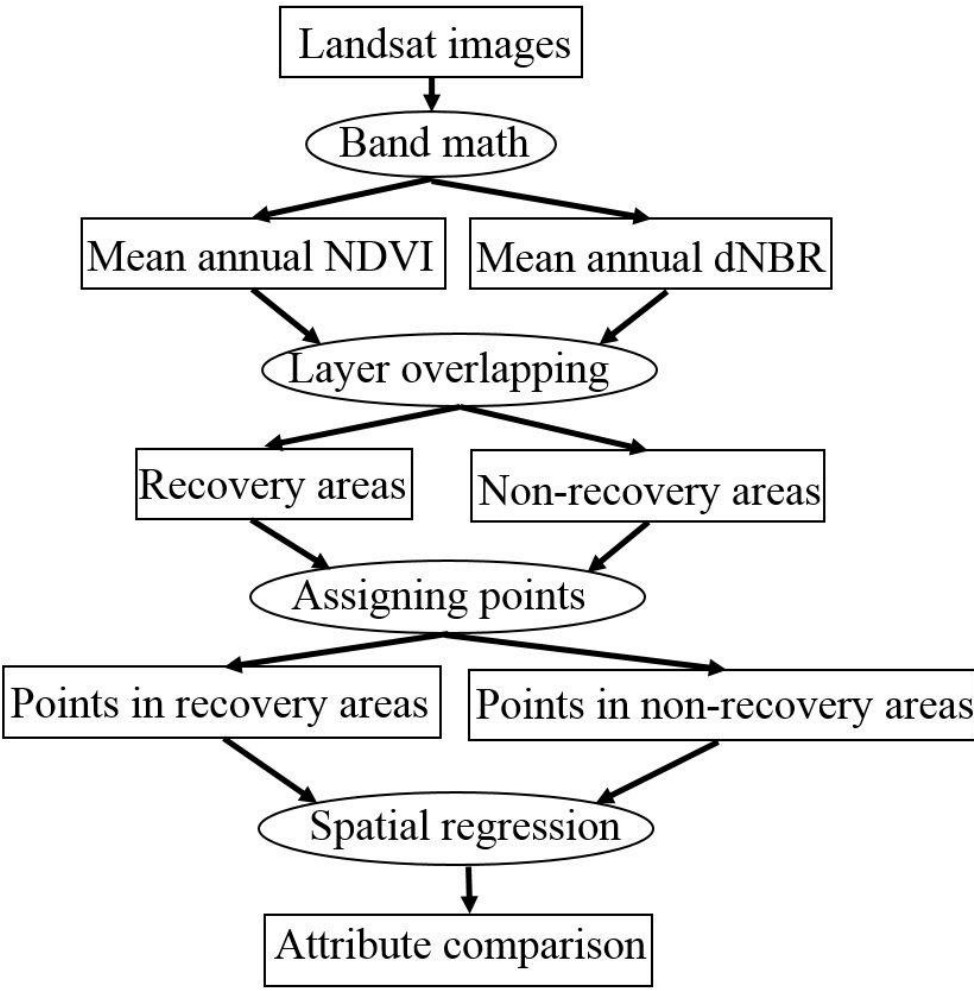

**Figure 2.** The workflow of our study.

All possible factors were classified into two types (Table 1): pre-fire drivers and post-fire characteristics. The pre-fire drivers are the possible factors leading to the non-recovery, which existed even before the 1988 fire and remained the same after the fire. The post-fire characteristics are the results of the interaction of the 1988 fire and pre-fire drivers. The pre-fire drivers included atmospheric temperature layers and precipitation layers, which were retrieved from the Natural Resources Conservation Service [22]; slope and elevation layers from the USGS; the pre-fire forest species in the Yellowstone National Park (https://irma. nps.gov/Datastore/Reference/Profile accessed on 18 May 2022) [23]; vegetation density (NDVI) from band computation among Landsat series; and soil order from Ramsey [21]. Post-fire characteristics, including bulk density, percentage of clay, percentage of silt, percentage of sand, pH, soil organic matter, total nitrogen, and magnesium (Mg), were from the Soil Properties and Class 100-m Grids United States Database (https://scholarsphere. psu.edu/resources/ea4b6c45--9eba-4b89-aba6-ff7246880fb1 accessed on 18 May 2022) [24] (Table 1). After the 1988 Yellowstone fire, the local forest agency applied a new management policy to ignite some prescribed fires deliberately to consume the accumulated fuel in the forest. Thus, soil characteristics in our study areas were relatively stable after the 1988 fire, and we classified them as post-fire characteristics. The survey for the soil properties was completed after the 1988 fire. We classified all soil properties into two types: stable soil properties and changeable soil properties. We assumed the stable properties of soil, such as soil order, would not change in a 30-year period or by a single disturbance. Thus, we classified soil order as a pre-fire driver. However, the changeable properties, such as pH,

bulk density, and some nutrients, could change in a short time period and by the 1988 fire, so we classified them as post-fire characteristics.

*2.3. Calculating dNBR and Their Thresholds*

We chose two Landsat scenes in the growing seasons of 1987 and 2018, respectively, to classify their distribution of land covers using the maximum likelihood method. There are six land covers: forest, grassland, barren land, water, cloud, and permanent snow. We also collected all available Landsat images in the growing seasons of 1989, 1998, 2008, and 2018 and computed the Normalized Burned Ratio (NBR) index for each date with Equation (1). We calculated median values among all NBR images for a specific year to generate an average NBR image for the growing season to avoid some random error and the influences of haze and clouds. We designated the 1987 NBR image as the pre-fire images and calculated the differenced Normalized Burn Ratio (dNBR) images for 1998, 2008, and 2018 with Equation (2).

$$\text{NBR} = (\text{Infrared Band} - \text{Mid-Infrared Band})/(\text{Infrared Band} + \text{Mid-Infrared Band}) \quad (1)$$

$$\text{dNBR} = (\text{pre-fire\_NBR} - \text{post-fire\_NBR}) \times 1000 \quad (2)$$

where Red Band, Infrared Band, and Mid-Infrared Band are the three bands in satellite images; pre-fire_NBR is the median NBR layer in 1987; post-fire_NBR layers include NBR layers in 1989 (right after the fire), 1998 (10 years after the fire), 2008 (20 years after fire), and 2018 (30 years after fire). We measured the difference between NBR pre-fire and NBR post-fire at each time point, respectively.

Each fire has its thresholds to define the burn severity class, and we tried to calculate the 1988 Yellowstone fire burn thresholds using dNBR and official fire maps [25] (Table 2, Figure 3A). The dNBR is one of the most widely used indexes to identify burn severity, although it could be affected by other disturbances. We used the burn severity map of the Yellowstone National Park from the Forest Service whose spatial resolution was 50 m for the ground truth information for the 1988 Yellowstone fire.

**Table 2.** Burn severity level and corresponding dNBR values.

| Severity Level | dNBR | Land Type |
|---|---|---|
| Unburned and Regrowth | −0.3000–0.1934 | Unburned |
| Moderate severity | 0.1935–0.4704 | Burned |
| High Severity | 0.4705–1.3500 | |

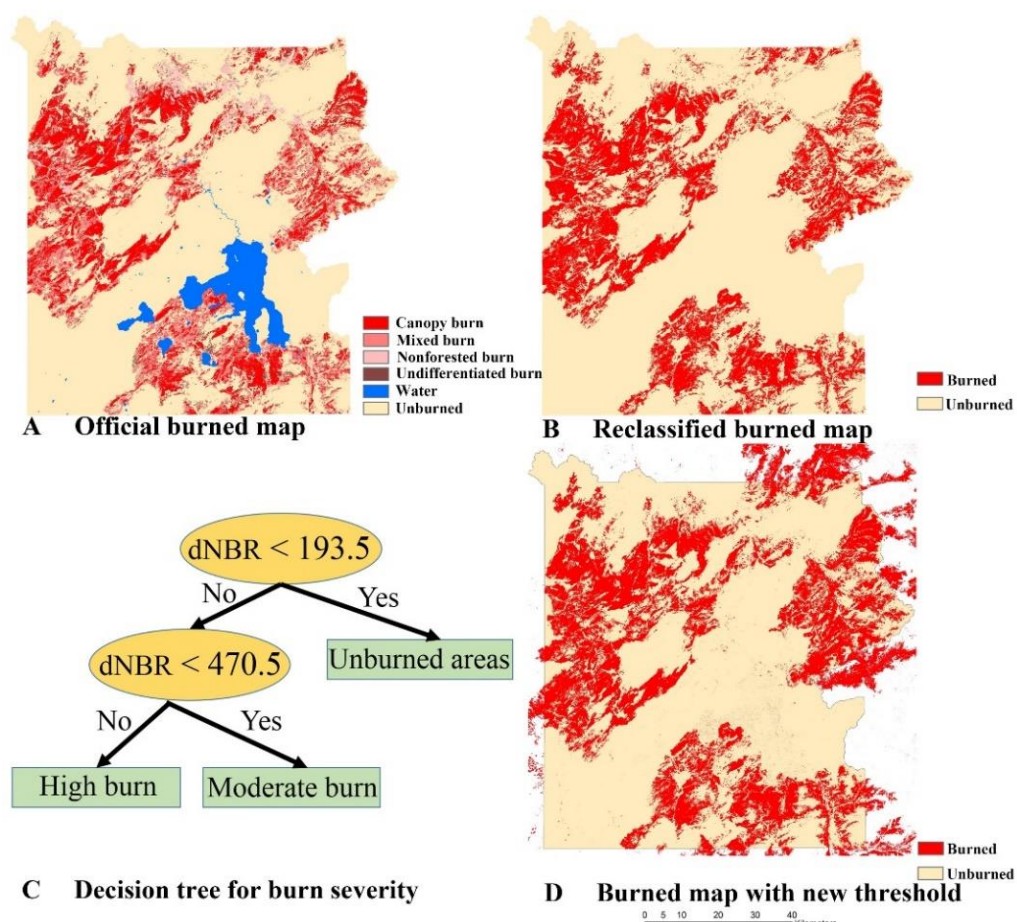

**Figure 3.** The comparison between the official burned map and the map using our decision tree.

We did not have the fire severity map for the whole study area, so we collected all training points from the park (a subset of our study area), generated the thresholds, and applied the rules to our whole study area. Then, the burn severity map was generated with the decision classifier (Figure 3).

With Key and Benson's [26] research, we reclassified the map with four burn severity classes. The canopy fire and the mixed fire in the legend were classified as high burn severity and moderate burn severity, respectively. The non-forest fire and undifferentiated fire were low burn severity, while the water and the unburned areas were classified as unburned areas. The definition of burning included the high burn, moderate burn, and low burn, as mapped in Figure 3B, based on the official map [25]. Within the national park, a training data set was built, where the interval of each grid was 200 m. The number of available training points was 222,395, and each training point had two attributes: burn severity from the severity map and dNBR value from the satellite images. The training set was put into the decision tree classifier to generate the thresholds for each burn severity in Figure 3C [27]. The number of low severity sites was small, so the classifier did not generate the burn severity for this class. There were three burn classes: Unburned areas, Moderate burn areas, and High burn areas (Figure 3C). Following the thresholds, all pixels were predicted within our study areas (Figure 3A). The spatial distribution of the burn severity in the National park using new thresholds is shown in Figure 3D with 84.8% accuracy.

### 2.4. The Extraction of Non-Recovery Areas

In our research, all four of the following requirements should be met to classify the pixel as a non-recovery area.

1. In the pre-fire period, land cover of the pixel should be classified as forest.

2. During the fire, the pixel was classified as a burned area.

3. In the post-fire period, land cover of the pixel should be classified as non-forest.

4. The pixel should keep the burned landscape at least as a moderate burn severity throughout our entire study time period.

According to the four requirements, we developed a model to classify the non-recovery areas (Equation (3)).

$$\text{Non-recovery} = \text{Pre-fire vegetation} \times \text{dNBR 1989} \times \text{Post-fire vegetation} \times \text{dNBR1998} \times \text{dNBR2008} \times \text{dNBR2018} \quad (3)$$

In order to meet the first requirement, the non-recovery pixel should be classified as forest in 1987, which is why we included 'Pre-fire vegetation' into the equation. To meet the second requirement, the dNBR value of the non-recovery areas should be categorized as a moderate burn or high burn, which is why we included 'dNBR 1989' into the equation. To meet the third requirement, the non-recovery pixels should be classified as grass or barren land in the 2018 land cover classification, which is why we assigned 'Post-fire vegetation' in the equation. To meet the fourth requirement, we checked the dNBR values in 1998, 2008, and 2018 to avoid the exception that other fires or droughts resulted in the non-forest in the 2018 classification but not the 1988 fire, which is why we put 'dNBR1998', 'dNBR2008', and 'dNBR2018' into the equation.

The burn areas should also meet the first two requirements of the non-recovery areas (Equation (4)).

$$\text{Burn area} = \text{Prefire vegetation} \times \text{dNBR 1989} \quad (4)$$

Recovery areas are the burn areas to the exclusion of the non-recovery areas (Equation (5)).

$$\text{Recovery areas} = \text{Burn areas} - \text{Non-recovery areas} \quad (5)$$

*2.5. Statistical Analysis*

There were two pools of possible drivers for non-recovery. One pool included the pre-fire drivers, which had a continuous influence on our study area before the 1988 Yellowstone fire and remained the same after the fire. This pool contains topographic factors, like elevation (m) and slope (degree); climatic factors, like precipitation (mm), maximum temperature (°F), and minimum temperature (°F); soil order layer; and pre-fire forest species layer. The second pool of drivers included the post-fire factors, which varied due to the fire, while also continuing to influence our study area post-fire. This pool consisted of soil physical properties from the Soil Properties and Class Database [21], like bulk density (kg/m$^3$), electrical conductivity (s/m), percentage of clay (%), percentage of silt (%), percentage of sand (%), and pH layers. The post-fire pools also included the layers that indicated soil chemical properties, like soil organic matter (g/kg), total nitrogen (mg/L), and the density of Mg (ppm). From the same soil database, our research selected the layers that were possibly relevant to forest recovery. All the layers represent topsoil with a depth of 0–5 cm. Topsoil is the only soil layer influenced by fire and is the only soil component that could be tested with satellite imagery.

In order to make the spatial distribution of the sample points even, we chose the systematic sampling pattern at an equidistant interval (150 m), such that there were 77,501 points for the recovery areas and 44,629 points for the non-recovery areas. With the same sample density for non-recovery and recovery areas, the method reflected what the distributions of these two types of areas were in the past. Our computer device could not perform this large-scale computation for 77,501 points and 44,269 points in the spatial regressions. Thus, we randomly selected one-tenth of the points (7750 points for recovery and 4426 for non-recovery) to perform the spatial regression. Then, we extracted attributes from the data layers for each selected point. In order to find the factors that influenced the post-fire recovery, we set the dNBR in 2018 as a dependent variable. Conversely, elevation, slope, maximum temperature (t_max), minimum temperature (t_min), annual precipitation (pre_a), Magnesium (Mg), Potassium (K), bulk density, electrical conductivity, sand percentage, clay percentage, silt percentage, soil organic matter, and prefire NDVI were

regarded as the independent variables. To determine the factors with a significant influence on recovery after fire, we went through three steps.

Firstly, the possible factors were chosen without consideration of collinearity and spatial autocorrelation. Secondly, the collinearities of possible factors were checked. Lastly, the spatial autocorrelations of possible factors were checked.

The specific approach was the following: All variables were input into a general linear regression in SPSS 28.0 [28]. We chose stepwise regression and computed the Variance Inflation Factor (VIF) to verify the collinearity. According to O'brien [29]'s research, if the Variance Inflation Factor is less than three, there is no collinearity. If the Variance Inflation Factor is between three and five, it is likely that collinearity exists. Factors with collinearity were excluded from the model, and the rest of the possible factors were input into a geo-spatial software, Geoda. The Moran I index was computed to test whether there was spatial autocorrelation. If spatial effects existed, we chose spatial error regression and spatial lag regression using a *p*-value of 0.05. If there was no spatial effect, we used multiple regression without spatial components. Finally, factors affecting the post-fire recovery were selected. Because the soil order layer and pre-fire forest species layers were categorical data, Chi-Square Tests were used to examine the difference between these two groups. The soil, climatic, and topographic properties were calculated according to different tree species in the national park to verify whether covariance with species existed.

## 3. Results

### 3.1. The Spatial Distribution of Non-Recovery Areas

The distribution of the non-recovery areas was sparse, most of which were concentrated in the high elevations of the study area. Four dNBR layers for 1989, 1998, 2008, and 2018 (Table 3, Figure 4) showed that the burn areas greatly decreased through time, with most of the recovery occurring in the first 10 years. The high severity areas and the moderate severity areas reached 2185.83 km$^2$ and 2108.85 km$^2$ in 1989, respectively, which were the highest among the five time steps. From 1989 to 1998, the high severity areas dramatically decreased from 2186.83 km$^2$ to 204.43 km$^2$. From 1998 to 2018, the areas of high severity burn fluctuated because there were new wildfires or prescribed fires in our study area causing new burns. We overlaid the land cover map and dNBR layers and extracted non-recovery areas, which covered 1005.25 km$^2$ and took up 3.42% of the whole study area (Figure 4).

**Table 3.** The areas for each burn severity from 1989 to 2018 (Unit: km$^2$).

| Burn Severity | 1989 | 1998 | 2008 | 2018 |
|:---:|:---:|:---:|:---:|:---:|
| High severity | 2185.83 | 204.43 | 304.59 | 120.07 |
| Moderate severity | 2108.85 | 3125.45 | 3513.36 | 7316.63 |
| Unburned or regrowth | 25,030.72 | 25,690.72 | 25,504.85 | 21,930.45 |
| No data | 102.04 | 406.83 | 104.63 | 60.28 |

Note: The occurrence of no data were outliers due to their dNBR values going beyond the valid range from −0.55 to 1.35.

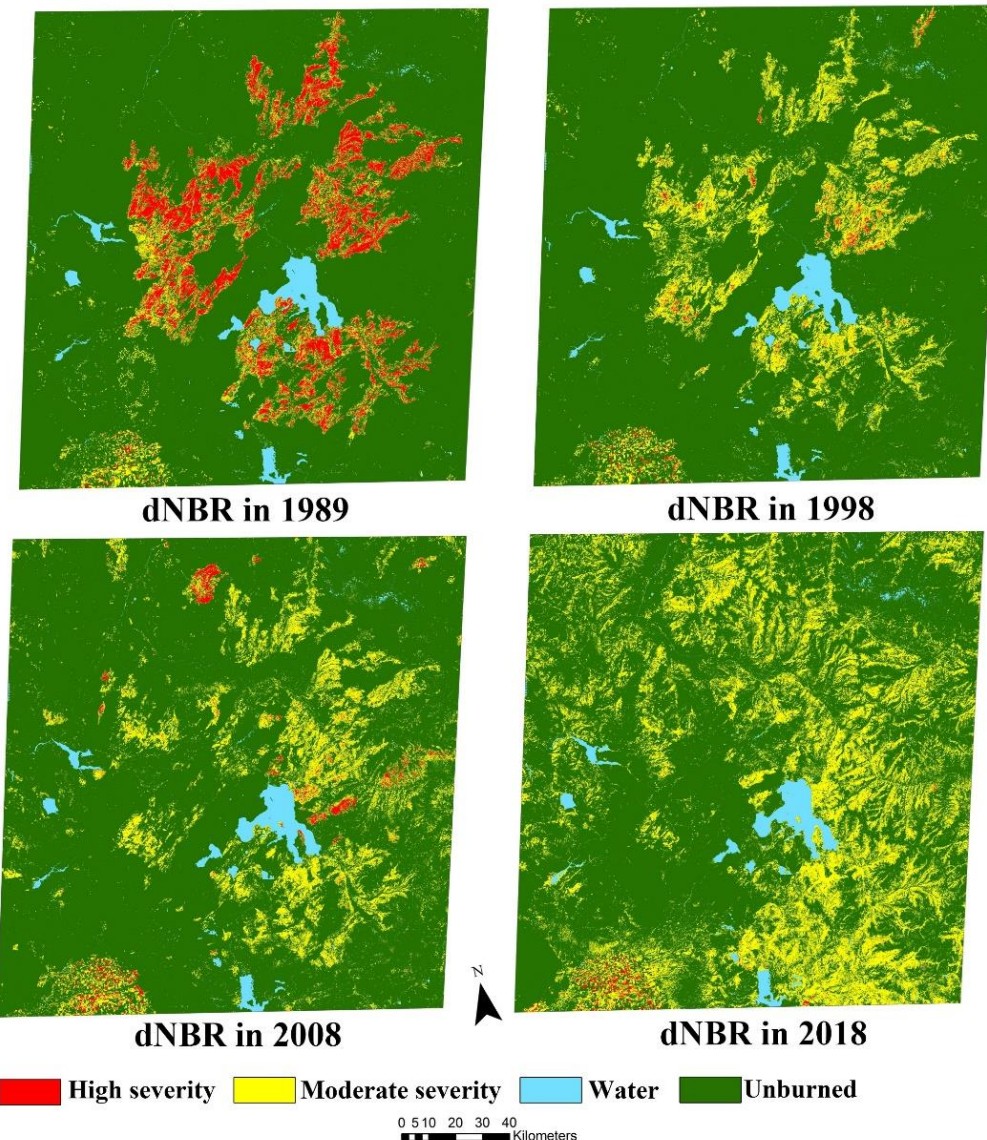

**Figure 4.** The burn severity maps from 1988 to 2018. The dominant tree species in the national park are aspen, Douglas fir, lodgepole pine, subalpine fir, and whitebark pine. Each species has its own ecological niche and species tolerances (Table 4). The whitebark pine lived in the higher elevation (2740.67 m), while Aspen grew in the lower elevation (2103.32 m). The whitebark pine had strong survivability, which occurred in the areas with a steep slope (16.99°), lack of organic material (189.94 g/kg), and low Nitrogen (60.65 mg/L). Aspen needed low precipitation (576.83 mm) and a high Nitrogen (76.69 mg/L) environment. Subalpine fir preferred acidic soils (pH = 5.66), and lodgepole pine occurred in areas of high organic matter (218.42 g/kg).

**Table 4.** Ecological niches for the dominant species in the national park.

|  | Elevation | Slope | Mean_Pre | Sand | pH | K | Organic Carbon | N |
|---|---|---|---|---|---|---|---|---|
| Aspen | 2103.32 | 10.06 | 22.71 | 40.83 | 6.36 | 168.04 | 188.52 | 76.69 |
| Douglas | 2213.18 | 19.15 | 22.63 | 44.33 | 6.22 | 179.72 | 217.86 | 67.63 |
| Lodgepole | 2422.60 | 8.03 | 33.33 | 43.70 | 5.72 | 176.71 | 218.42 | 61.13 |
| Subalpine | 2524.73 | 10.24 | 38.88 | 40.65 | 5.66 | 176.17 | 202.39 | 61.08 |
| Whitebark | 2740.67 | 16.99 | 37.82 | 44.53 | 5.72 | 171.20 | 189.94 | 60.65 |

Note: Mean_Pre is the annual mean precipitation.

### 3.2. The Differences between Non-Recovery Areas and Recovery Areas

The multiple regression analyzed the quantitative factors (Table 5), which chose 11 variables as potential factors (elevation, prefire NDVI, Soil organic, annual precipitation, pH, t_max, total of nitrogen, K, t_min, sand, and slope) for recovery areas and 13 variables as potential factors (DEM, prefire NDVI, Soil organic, annual precipitation, pH, t_max, total of nitrogen, K, slope, silt, bulk density, and Mg) for non-recovery areas. Among potential factors for recovery areas, elevation and t_max were collinear, with high Variance Inflation Factors (4.73 and 4.72), larger than three, whose correlation (r) was $-0.86$. The t_max was excluded because elevation was a more important factor than the t_max. Among potential factors for non-recovery areas, elevation and t_max also had high Variance Inflation Factors (3.74 and 3.38), larger than three, whose correlation was $-0.80$. The t_max was also excluded.

**Table 5.** The chosen factors for the recovery and non-recovery using general regression.

| | Recovery | | | Non-Recovery | | |
|---|---|---|---|---|---|---|
| | Coefficient | *p* | VIF | Coefficient | *p* | VIF |
| constant | 954.58 | <0.01 | - | 618.94 | <0.01 | - |
| elevation | −0.15 | <0.01 | 4.73 | −0.10 | <0.01 | 3.74 |
| pre_NDVI | 0.40 | <0.01 | 1.08 | 0.45 | <0.01 | 1.07 |
| soil organic | 0.27 | <0.01 | 1.44 | −0.10 | <0.01 | 1.20 |
| annual precipitation | −2.73 | <0.01 | 2.07 | −0.66 | <0.01 | 2.40 |
| pH | −3.31 | <0.01 | 1.98 | −1.46 | <0.01 | 1.96 |
| t_max | −5.91 | <0.01 | 4.72 | −1.82 | 0.00 | 3.38 |
| total of nitrogen | −0.28 | <0.01 | 1.07 | −0.09 | <0.01 | 1.213 |
| K | −0.19 | <0.01 | 1.17 | −0.41 | <0.01 | 1.13 |
| t_min | 2.81 | <0.01 | 1.35 | - | <0.01 | - |
| sand | −0.53 | <0.01 | 1.33 | - | <0.01 | - |
| slope | −0.35 | <0.01 | 1.21 | −0.46 | <0.01 | 1.21 |
| silt | - | - | - | 0.50 | <0.01 | 1.15 |
| bulk density | - | - | - | −0.03 | <0.01 | 1.62 |
| Mg | - | - | - | 0.00 | 0.01 | 1.38 |

Note: "-" indicates that the results of that item did not meet significant levels. VIF is the Variance Inflation Factor.

The Moran's I in both areas (recovery areas and non-recovery areas) were 5.31 and 29.04 (Table 6), whose *p*-values were below 0.01, which indicates that both areas had obvious spatial autocorrelations.

**Table 6.** The spatial check for recovery areas and non-recovery areas.

| | Recovery | | | Non-Recovery | | |
|---|---|---|---|---|---|---|
| TEST | MI/DF | Value | Prob | MI/DF | Value | Prob |
| Moran's I (error) | 0.00 | 5.31 | <0.01 | 0.02 | 29.05 | <0.01 |
| Lagrange Multiplier (lag) | 1.00 | 2.63 | 0.11 | 1.00 | 59.97 | <0.01 |
| Robust LM (lag) | 1.00 | 9.59 | <0.01 | 1.00 | 3.29 | 0.07 |
| Lagrange Multiplier (error) | 1.00 | 15.90 | <0.01 | 1.00 | 346.89 | <0.01 |
| Robust LM (error) | 1.00 | 22.85 | <0.01 | 1.00 | 290.21 | <0.01 |

Note: Moran's I determined whether there was a spatial effect in the areas.

As the *p*-value for lag regression in recovery areas was 0.11 and the *p*-value for the robustness of lag regression in non-recovery areas was 0.07, we excluded the lag regression. The spatial error regressions were applied to recovery and non-recovery areas (Table 7). The dependent variable was the dNBR in 2018, where higher values indicated higher burn severity. The mean dNBRs in recovery areas and non-recovery areas were 0.21 and 0.26. Elevation, annual precipitation, spatial effect, pH, and K all showed negative effects on both areas, while pre-fire NDVI showed positive effects. Elevation, annual precipitation, and pH in the non-recovery areas were higher (elevation = 2568.38 m, annual precipitation = 34.32 in

(871.72 mm), pH = 5.94) than those in the recovery areas (elevation = 2440.50 m, annual precipitation = 33.57 in (852.68 mm), pH = 5.84). Many factors influenced both areas, but few factors affected either recovery areas or non-recovery areas. The percentage of sand and minimum temperature only affected recovery areas, while the percentage of silt, bulk density, and Mg only affected the non-recovery areas. Some factors, like slope and soil organic matter, showed opposite effects on the recovery areas and non-recovery areas.

**Table 7.** The spatial error regression for recovery areas and non-recovery areas.

| | Recovery | | | Non-Recovery | | |
|---|---|---|---|---|---|---|
| | **Coefficience** | ***p*** | **Average** | **Coefficience** | ***p*** | **Average** |
| Constant | 731.89 | <0.01 | - | 910.37 | <0.01 | - |
| Elevation | –0.07 | <0.01 | 2440.50 | –0.17 | <0.01 | 2568.38 |
| Pre_NDVI | 0.31 | <0.01 | 0.37 | 0.77 | <0.01 | 0.38 |
| Soil organic | 0.05 | <0.01 | 196.95 | –0.34 | <0.01 | 165.61 |
| Annual precipitation | –3.72 | <0.01 | 33.57 | –1.22 | <0.01 | 34.32 |
| pH | –5.45 | <0.01 | 5.84 | –11.36 | <0.01 | 5.94 |
| Total of nitrogen | –0.63 | <0.01 | 63.50 | 0.34 | <0.01 | 60.97 |
| K | –0.24 | <0.01 | 172.47 | –0.66 | <0.01 | 171.96 |
| t_min | 5.02 | <0.01 | 22.60 | - | - | 22.54 |
| sand | –1.68 | <0.01 | 44.21 | - | - | 46.56 |
| slope | –0.56 | <0.01 | 11.26 | 1.81 | <0.01 | 13.99 |
| Spatial effect | –4.05 | <0.01 | - | –34.07 | <0.01 | - |
| Silt | - | - | 41.5 | 3.07 | <0.01 | 39.07 |
| Bulk density | - | - | 715.92 | 0.19 | <0.01 | 750.03 |
| Mg | - | - | 8443.10 | 0.01 | <0.01 | 10,435.46 |

Note: "-" indicates that the results of that item did not meet a significant level.

The Chi-square focused on qualitative factors, like soil order and pre-fire forest species (Table 8). The pre-fire forest species ($p < 0.001$) and soil order ($p < 0.001$) in the non-recovery areas and recovery areas were also statistically different in the Chi-square analysis (Table 8). Inceptisols made up 50% of the non-recovery areas and 51% of the recovery areas. Mollisols took up 14% of the non-recovery areas and 11% of the recovery area (Figure 5). Alfisols took up 23% of the non-recovery areas and 18% of the recovery areas.

**Table 8.** Chi-square analysis between non-recovery and recovery areas.

| | **Pearson Chi-Square** | **df** | **Asymp. Sig. (2-Sided)** |
|---|---|---|---|
| Soil order | 2694.67 | 5 | <0.01 |
| Tree species | 6201.16 | 7 | <0.01 |

Note: Soil orders in our study included Alfisols, Aridisols, Entisols, Mollisols, Inceptisols, Mollisols, and no soil. The forest cover included Aspen, Douglas, Lodgepole, Subalpine Fir, Water, and Non-forest.

The dominant tree type for both non-recovery and recovery areas in the pre-fire period was lodgepole pine (61% and 87%; Figure 6). However, whitebark pine was much more common in the non-recovery areas (non-recovery: 24%, recovery: 7%), and subalpine fir was also more likely to occur in the non-recovery areas (non-recovery: 11%, recovery: less than 1%). The NDVI and burn severity (dNBR between 1987 and 1989) in the non-recovery (NDVI = 0.38; dNBR = 643.44) were higher than that in the recovery areas (NDVI = 0.37; dNBR = 476.52).

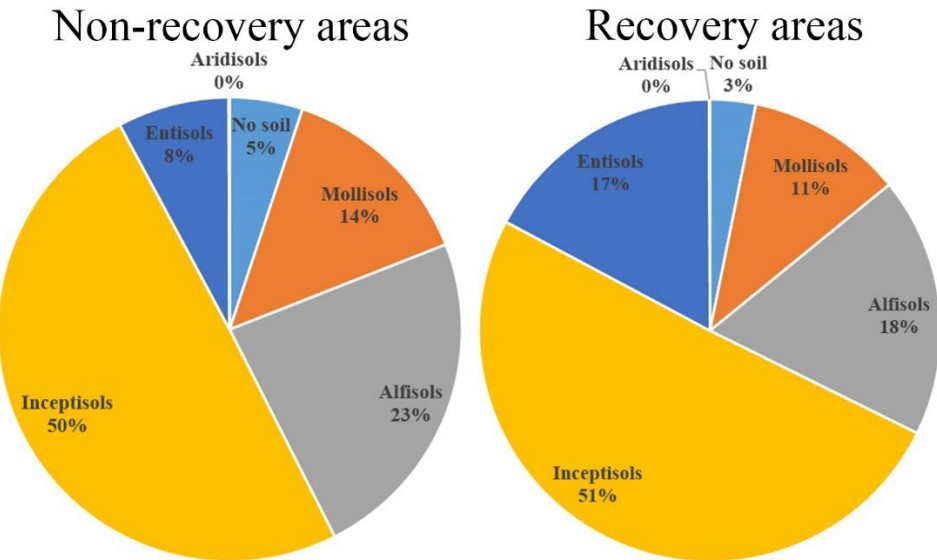

**Figure 5.** Soil compositions for non-recovery areas and recovery areas.

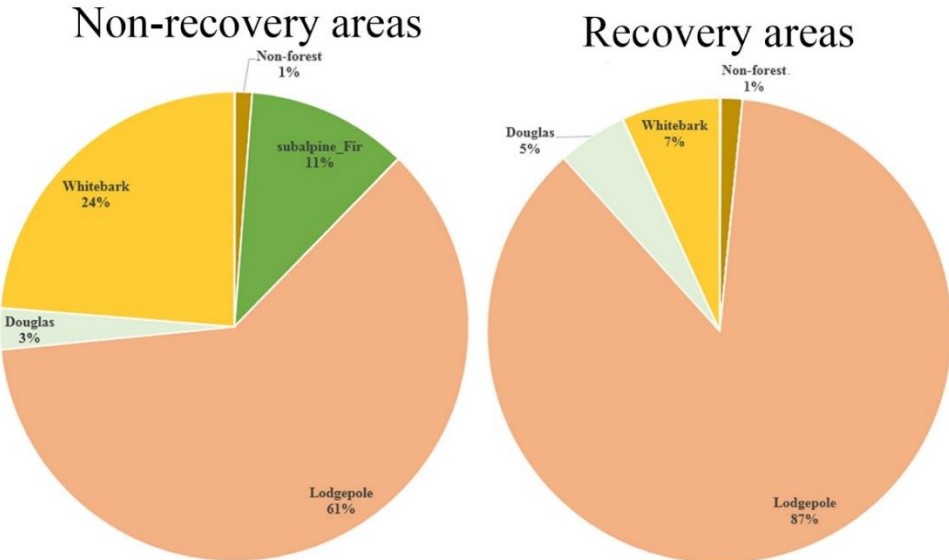

**Figure 6.** Pre-fire forest species for non-recovery areas and recovery areas.

## 4. Discussion

From the land cover analysis, most forest areas recovered in 10 years, as demonstrated by the reduction of high severity areas from 2185.83 km² in 1989 to 204.43 km² in 1998. Peterson [30] considered fire as a keystone process to drive the ecosystem dynamics by producing persistent landscape vegetation patterns. Based on their previous ecological memory, quick recovery verifies the resilience of these forest communities, whose dominant species were conifer trees [31]. From the large changes between pre-fire and post-fire, the results showed the influences of the pre-fire drivers and the post-fire characteristics (Table 3). Pre-fire drivers such as climatic factors directly influence the recovery, while a higher pre-fire NDVI indirectly affects the recovery via high severity burns, while the post-fire characteristics like soil nutrients could affect the recovery directly. These factors could exert direct effects on tree growth or might be covariate factors that indirectly influence recovery by limiting tree establishment.

*4.1. Effects of Pre-Fire Drivers on Recovery and Non-Recovery Areas*

The pre-fire drivers (information legacies) directly played a significant role in forest recruitment. The pre-fire drivers involved in our study are long-lasting and unchanged in the post-fire period, including topographic factors (elevation and slope); climatic factors (minimum temperature and annual precipitation); soil order; and biotic drivers, like pre-fire forest species and pre-fire forest vegetation density. From the regression analysis (Table 7), non-recovery areas had higher elevation (2568.38 m) and steeper slopes (13.99°) than those in the recovery areas. We compared the ecological niche for each species (Table 4) with the characteristics for non-recovery and recovery areas (Table 7). The average elevation for the non-recovery (2568.38 m) was higher than the average elevation for most of the species, such as aspen, Douglas fir, lodgepole pine, and subalpine fir. Elevation could be a covariance with tree species in areas where elevation is a limiting factor for the tree growth for specific species, which then indirectly influences the recovery in the post-fire period. A steeper slope in the non-recovery areas could reduce the amount of moisture retained in the soil and drive more soil erosion. Annual precipitation in the non-recovery areas (852.68 mm) was higher than that in the recovery areas (871.73 mm), which could also drive soil erosion on steep slopes, which limits tree growth. Accelerated erosion after wildfire is a major form of land degradation [32]. Besides the direct effects to the landscape, erosion is a threat for species survival and growth, even within environmentally protected areas [32]. Soil biodiversity, like soil organisms, also contributed to the soil loss in the post-fire period [33]. From soil order distribution, this study found a higher percentage of Mollisols (14%) and Alfisols (23%) in the non-recovery areas than in the recovery areas (11% for Mollisols and 18% for Alfisols). Abella et al. [34] stated that the Mollisols were associated with grassland and the Alfisols with forest, which indicated that the ecological memory of a site was important, with some of the area reverting to grassland.

Trees have adapted to and have evolved through many disturbances, such that the characteristics of the disturbance regimes differ in different forest types [35]. From the forest species distribution in the National park (Figure 6), the dominant tree species had various fire-resistant abilities. According to the corresponding research [36], the ranking of fire resistance for the dominant tree species is Douglas fir (very high fire resistance) > whitebark pine = lodgepole pine (low fire resistance) > subalpine fir (very low). The non-recovery areas had a lower percentage of Douglas fir (3%) and a higher percentage of subalpine fir (11%) than those in the recovery areas (5% for Douglas fir and less than 1% for subalpine fir). Tree species with low fire resistance are more susceptible to fire and contribute to accelerating the breakdown of the previous landscape structure. Peterson [30] highlighted the role of the ecological memory relationship between fire and landscape patterns, such that fire alters landscape patterns, but the landscape pattern (forest type patches), in turn, governs the extent of fire if there is ecological memory. The biotic pre-fire drivers, such as tree species and NDVI, led to the higher burn severity in the non-recovery areas, which slowed down the post-fire recovery.

The growth of tree saplings requires substantial water and nutrients. Higher annual precipitation in the non-recovery areas could wash away the nutrients as runoff. In addition, soil can rarely hold water on a steeper slope, even though the precipitation was higher in the non-recovery areas. The regeneration of forest demands higher nutrients and water for growth than grassland, but the non-recovery areas had fewer nutrients and water.

*4.2. Effects of Post-Fire Characteristics on the Recovery Areas and Non-Recovery Areas*

Most of the post-fire characteristics (material legacies) caused by the fire and pre-fire drivers could also affect the recovery of the areas. The post-fire characteristics collected in the post-fire period from Table 1 in our study mainly focused on the change of soil physical and chemical characteristics. The change of soil physical characteristics was caused by the alteration of the porosity between soil granules. The burned ash caused by more severe fire and more precipitation in the non-recovery areas can reduce porosity. The high water repellency led to drier micro-environments in the non-recovery areas, and without good

water-holding capacity, mudslides occurred due to heavy rain. The higher bulk density in the non-recovery areas (750.03 kg/m$^3$) than that in the recovery areas (715.92 kg/m$^3$) after fire indicates soil compaction, which may have impeded seedling growth and limited root growth [37]. The higher percentage of sand (46.56%) and the lower percentage of silt (39.07%) increased the loss of water and nutrients.

Post-fire chemical characteristics (like unsuitable soil acidity and lack of soil nutrients) may lead to non-recovery areas. Compared with grass or shrubs, trees need more nutrients and water absorbed by shallow roots. The pH increases contribute to the increased availability of cations after a fire and the loss of organic acids during oxidation of litter and organic matter [38]. The local dominant vegetation was pine, which grows on acidic and well-drained soil and prefers a soil pH from 4.5 to 6 [39]. However, the average pH values were 5.84 and 5.94, respectively, in the recovery areas and the non-recovery areas. Compared to the pines, grassland with an optimum pH value around 6.5 was more likely to regenerate in the non-recovery areas [40]. One of the significant factors of the non-recovery areas, Mg concentration (10435.46 ppm in the non-recovery areas and 8443.10 ppm in the recovery areas), could lead to high pH values. High severity fire can volatilize organic carbon in the soil and decompose nitrogen-rich organic matter into inorganic composites like ammonium ($NH_4^+$) and nitrate ($NO_3^-$) [41]. Compared with organic nitrogen composites, $NH_4^+$ and $NO_3^-$ are easily absorbed by the plants. However, the existence of $NH_4^+$ cations is ephemeral and easily leached deeper into the soil, especially by higher precipitation in the non-recovery areas, which could explain the lower organic density and lower nitrogen density in the non-recovery sites [42]. Tree regeneration takes more time than the grass. When the tree saplings try to absorb nitrogen, $NH_4^+$ could have been carried away by the runoff. The minimum requirement of soil organic matter for all dominant species was 170.90 g/kg (Table 4). However, the organic material in the non-recovery areas was 165.61 g/kg (Table 6). None of the tree species are likely to grow in infertile areas with poor organic matter. The average nitrogen value for the non-recovery areas was 60.97 mg/L, and only whitebark pine (60.65 mg/L) could adapt to this environment. The values here were the average values, and there might be some trees growing in those areas, but the number of extreme cases was small. The difference in the values of organic matter between the non-recovery and recovery areas was 31.16 g/kg, which possibly caused limited regrowth in the non-recovery areas. Organic matter and nitrogen could be the most important factors that impact the recovery in the post-fire period.

### 4.3. Personal Observation on Non-Recovery Areas

Even though the total area of non-recovery areas was 1005.25 km$^2$, these areas are prone to natural disasters, which could affect human establishments. When the non-recovery areas are on steep slopes, landslides and mudflows can occur, where soil gravity outweighs the resistance to slide, such as friction, soil cohesion, and rooting strength (Figure 7).

Previous research [43] has demonstrated that after the 1988 wildfire, some non-recovery areas could experience mudflows without the stability from deeply rooted trees. As Figure 1 indicates, the pattern of the non-recovered areas was distributed sparsely and mixed with recovered areas; more work needs to be done to understand other possible follow-on disturbances, such as bark beetle outbreak, local drought, and wind effects, which can cause cascading effects on forest development and recovery. Whether fire exerts an amplifying or buffering interaction with follow-on disturbances needs more research [44]. The reestablishment in the recovery areas was also significant in the local ecosystem. The seed for reestablishment in the post-fire areas had two sources [11]: One was long-distance transportation, like the seed dispersal by wind, water, or animal. This type of seed usually has a light mass, and its dispersal efficiency has a negative correlation with the distance to the seed sources [45]. The more efficient way to spread seed is from the surviving trees or pine cones. For example, the lodgepole pine cones cannot open until they experience high-

intensity fires, and the closed serotinous pine cones can live for more than 40 years. Douglas fir had thick bark, which could survive surface fire and spread seeds in the post-fire period.

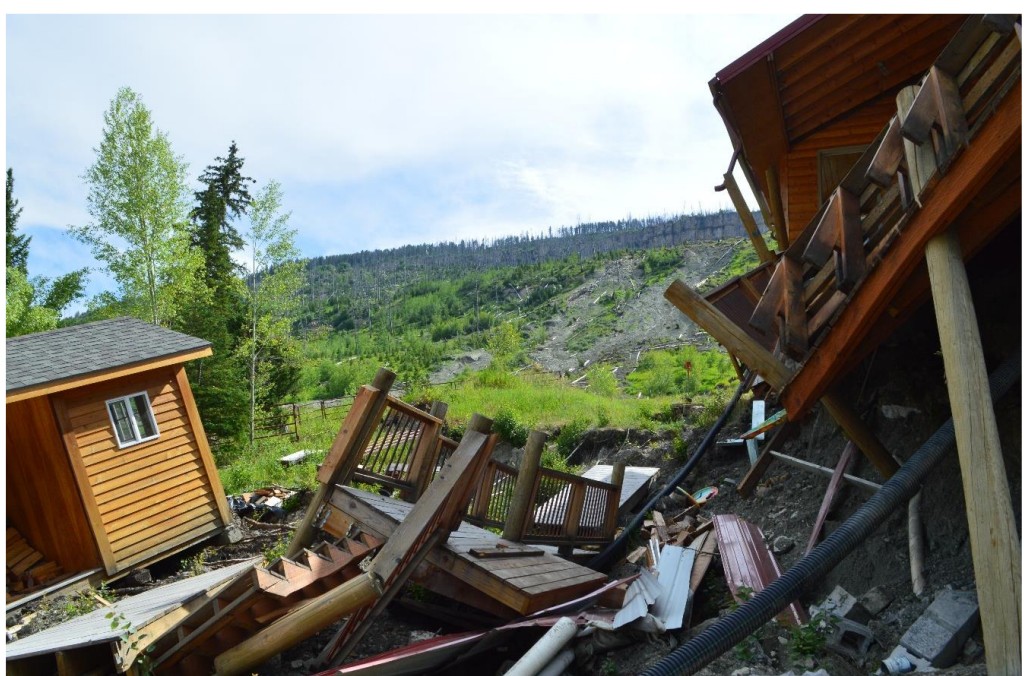

**Figure 7.** The mudslide occurred in one of the non-recovery areas, resulting in damage to multiple houses.

*4.4. Future Work*

The area of the GYE affected by the 1988 wildlife is enormous and has complex landscapes, such as valleys, plateaus, and mountains. Given the diverse and complex landscape in GYE, the results and assumptions made from this study might not match each sub-region in the GYE, and some areas experienced multiple fire disturbances, which can be explored in future analyses. Another issue is that the coarse resolution images could not identify the tree species accurately. More fieldwork can be done in the future studies to develop ground truth data, like marking some tree species in our study areas to verify results from our current research. The assessment of wildfires' footprint in the regulating ecosystem services could also be a target for future research.

**5. Conclusions**

Most areas in the GYE recovered, and their ecological memories successfully brought their forest community back within 10 years of the 1988 fire. This study found that the high severity burn areas were 204.43km$^2$ in 1998, and the non-recovery area was 1005.25 km$^2$, which triggered follow-on disturbances, like mudflows, and caused damage and loss of properties. The hypothesis of this study was verified, showing that the soil characteristics, pre-fire tree species, and topographic variables were different between the recovery areas and non-recovery areas. The non-recovery areas had higher elevation, lower annual temperature, higher annual precipitation, less soil organic, less Nitrogen, more subalpine fir, and more whitebark pine in the pre-fire period. Elevation was a significant covariant factor, which was identified as a driver of non-recovery and also controlled the tree species that originally grew in the non-recovery areas.

The study also found that the pre-fire density and soil nutrients (organic matter and Nitrogen) might play a significant role in burn recovery. Soil order also affected fire intensity and indirectly influenced recovery. The percentage of Mollisols was found to be higher in the non-recovery areas than the recovery areas. The hypothesis about the soil characteristics was also verified with the statistical differences for the post-fire characteristics (i.e., Organic

carbon and Nitrogen) between non-recovery and recovery areas, affecting the foundation of the new ecological system.

Even though the areas of non-recovery caused by the 1988 fire were small, their existence could potentially interact with other disturbances, like mudslides, and the compound effect may enhance the damage caused by wildfires. The drivers found in the study, like the topographic factors, climatic factors, biotic factors, soil properties, and fire memory, could interact with each other and maintain the new state of the ecosystem with new legacy memories. The non-recovery areas were caused by direct and indirect factors. The changed soil properties (material legacy), like organic material and Nitrogen, were direct factors that limited tree growth. The topographic effect (information legacy), as a covariant and indirect factor, selected the dominant tree species in the pre-fire period (higher percentages of subalpine fir and whitebark pine) and then indirectly determined vegetation recovery in the post-disturbance. The ecological memories consisted of pre-fire drivers and post-fire characteristics maintaining the dynamic equilibrium in the GYE. The current boundary between recovery and non-recovery areas will remain stable until the next disturbance, such as fire, drought, or severe climate change, perturbs the existing system.

**Author Contributions:** Conceptualization, H.L. and J.H.S.; methodology, H.L. and I.T.; formal analysis, H.L.; writing—original draft preparation, H.L. and I.T.; writing—review and editing, J.H.S. All authors have read and agreed to the published version of the manuscript.

**Funding:** Our experiences at the North American Dendroecological Fieldweek (NADEF) helped inform this analysis, and that fieldweek was funded by an NSF grant (BCS-1759694).

**Institutional Review Board Statement:** Not applicable.

**Informed Consent Statement:** Not applicable.

**Data Availability Statement:** The datasets for tree-ring, remote sensing, and statistical analysis are stored in the drive of the corresponding author, Department of Earth and Environmental Systems, Indiana State University, Terre Haute, IN 47809, USA.

**Conflicts of Interest:** The authors declare no conflict of interest.

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
