# Peer review of "Analyzing Resilience in the Greater Yellowstone Ecosystem after the 1988 Wildfire in the Western U.S. Using Remote Sensing and Soil Database"

_land, doi:10.3390/land11081172_

Round 1
Reviewer 1 Report
Analyzing resilience in the greater Yellowstone ecosystem after the 1988 wildfire using remote sensing and soils
Dear Authors
The basic science of this paper is conducted in a good way and is of appropriate standard. The author and his team write this paper according to journal scope and modern trends. I am glad to review this paper but there is no novelty in this paper. I have seen many papers related this topic and study area has been published in well reputed journals. If authors want to publish this study, they should provide some novelty or enhance the significance of the research. Moreover, paper is well-structured. I am going to recommend minor changes this paper. I hope author will follow our comments and enhance own study and resubmit again in this journal.
Minor comments
Title
•Title is fine and according to the study
Abstract
•Author tries to write in a better way but still, there are some mistakes. The author should follow the content of abstract. Abstract is very rigorous. Author should improve this part
• Background
• Context
• Objectives
• Material and methods
• Results/findings
• Novelty and purposes of this research
· Abstract section is very small. Provide the back ground then start objective of the research.
· Provide some quantitative results instead of theocratical.
· At the end of the abstract provide what is significance of this research.
Introduction
In this introduction section, there are so many reference problems. The author should check all references accordingly to the context.
I found many type errors in the whole manuscript.
If the author discusses the background, he/she should use old references. Why author used latest references here. All these are not suitable.
There is some typo error. The author should double-check the whole manuscript.
Objectives are not clear at the end of the introduction part
I believe the authors can demonstrate this and it should only take a couple of paragraphs in the introduction and discussion to show it but it is very important to do so if they wish to publish in an international journal.
Material and Methods
Ø First of all, the author should split the site description and data collection
Ø Add coordinates of study area.
Ø (Yellowstone National Park and sections of the Sho shone National Forest totaling) Where this park exits, add latitude and longitude in the study area.
Ø In data source, the author should explain how to obtain this data.
Results
I am agreed of results section
Discussion
Agreed
Conclusion
Conclusion section is very lengthy and very theocratical. The author should write some quantitative results in the conclusion section.
The author should revise the conclusion section
I hope authors will improve this study and resubmit again in this journal.
Best Regards
Reviewer 2 Report
This is interesting research. However, this manuscript should be paid more attention to revision and improvement before publication.
1. The Abstract need to supply more results and the meaning of the conclusion.
2. lines 56-57, the land use type change before and after the fire could reduce which influence on ecosystem structure/function? It should be more clear.
3. I can’t see the specific scientific problem, and can’t find the straightforward connection between the introduction and your hypothesis.
4. Supplement the reasons for the selection of general factors, how to consider the different resolutions of different data resources?
5. The result showed that elevation is the major factors effect the recovery. However, there lack more detailed and deeper analysis and discussiton to illustrate it.
Reviewer 3 Report
The article deals with the resilience on Yellowstone ecosystem after wildfire.
The title is not very informative. What do you mean ….soils? maybe sample plot? Also, I do not know (as probably many of the readers) where Yellowstone ecosystem is, please add the location in the tile.
Line 15. Avoid using “we found” it is more preferable to write “the analysis revealed”, “the results showed” ect. Please check and revise throughout the text as such expressions appeared many times.
Line 139. Please give the classification and limits of dNBR values for the characterization of burn severity.
In the last paragraph of the introduction the novelty points of the current approach must be highlighted in comparison with other similar research to the best author’s knowledge. State the research gap answered from this research.
Line 365. Accelerated erosion after wildfire is a major form of land degradation (Stefanidis et al., 2022). Asides the direct effects to the landscape erosion is a threat for species survival and growth, even within environmental protected areas (Stefanidis et al., 2022)
Orgiazzi, A., & Panagos, P. (2018). Soil biodiversity and soil erosion: It is time to get married: Adding an earthworm factor to soil erosion modelling. Global Ecology and Biogeography, 27(10), 1155-1167.
Stefanidis, S., Alexandridis, V., & Ghosal, K. (2022). Assessment of Water-Induced Soil Erosion as a Threat to Natura 2000 Protected Areas in Crete Island, Greece. Sustainability, 14(5), 2738.
Line 458. Also, the assessment of wildfires footprint in the regulating ecosystem services could be a target for future research.
The article in the current form is really weak. However, the approach is very interesting. My suggestion in the aforementioned parts I believe will strengthen the sound of the research.
Reviewer 4 Report
Overall:
This paper was very interesting to read and was very well written. I think that the conclusions make sense and are somewhat self-evident, but having data to back up these assumptions is a very important part of science. Therefore, I feel that this paper does make a helpful contribution to fire science/ecology.
Line by line comments:
Line 30: the Yellowstone fire is mentioned here as if the reader should already know about it. You do a good job in the next paragraph setting the background of the fire. I think that this sentence needs to be reworded to possibly just reference extreme fire events in general, and not Yellowstone specifically.
Line 58: good justification for the potential need/ramifications of the study
Line 83: grammar/punctuation. Suggest: “A long-term drought accumulated deadwood and dried materials, providing ample fuel.”
Line 103: should this be medium or median? If it is medium, can you briefly explain what that is?
Line 113: Resources should be plural in NRCS
Figure 2: Figure is soft/blurry, I recommend re-saving it at a higher resolution
Line 136: spell out NBR on first use
Line 139: spell out dNBR on first use
Line 144: medium is used again, just making sure it is not ‘median’
Line 153: suggest change “just a part of” to “a subset of”
Line 187-195: had you thought of checking the Monitoring Trends in Burn Severity (MTBS) dataset? They have landsat-derived burn polygons for the continental US going back to the 1980s. could be useful to check for reburned areas on a finer scale. I am worried that using a 10-year interval to check for re-burns might be too far apart.
Line 222: “The current equipment” reads weird. Does this mean the equipment that you had available to work on? I would suggest rephrasing that somehow.
Line 253: suggest placing a comma after “sparse”
Figure 4: is there an explanation for the dramatic increase in moderate severity in 2018? Was there a lot of reburns between 2008 and 2018?
Line 287: I would guess that elevation and t_max would be colinear, so this would make sense
Line 292-293: I would suggest rewording since it is self-apparent that those values are larger than the p threshold
Line 301: suggest using metric units instead of inches
Table 4: It is generally not ideal to have a table break across the page, but with all the consecutive figures and text in this section, I don’t know how to fix this. You could move table 4 up to immediately after the paragraph that mentions it, and I think it would all fit on the page, but that will probably introduce weird page breaks with the rest of the tables and figures, so I will leave it up to you. Overall, I think that there should be more of a break between the tables and figures, but it is what it is.
Tables 4-6: since you are rounding to 2 decimals, I would suggest putting less than (<) on the P values that are less than 0.01, so it is not assumed that they are zero.
Table 7: Note says “my study” instead of “our study”
Line 471: Singular noun used with plural verb. Should either use hypothesis/was or hypotheses/were
Round 2
Reviewer 2 Report
The quality of the manuscript has been improved after revision, i recommend to publication.
Reviewer 3 Report
The authors adress all the reviewer comments
This manuscript is a resubmission of an earlier submission. The following is a list of the peer review reports and author responses from that submission.